# Stress Activated MAP Kinases and Cyclin-Dependent Kinase 5 Mediate Nuclear Translocation of Nrf2 via Hsp90α-Pin1-Dynein Motor Transport Machinery

**DOI:** 10.3390/antiox12020274

**Published:** 2023-01-26

**Authors:** Tetsuro Ishii, Eiji Warabi, Giovanni E. Mann

**Affiliations:** 1School of Medicine, University of Tsukuba, Tsukuba 305-8577, Japan; 2King’s British Heart Foundation Centre of Research Excellence, School of Cardiovascular and Metabolic Medicine & Sciences, Faculty of Life Sciences & Medicine, King’s College London, 150 Stamford Street, London SE1 9NH, UK

**Keywords:** Nrf2, Hsp90, ERK, JNK, Cdk5, Pin1, HO-1

## Abstract

Non-lethal low levels of oxidative stress leads to rapid activation of the transcription factor nuclear factor-E2-related factor 2 (Nrf2), which upregulates the expression of genes important for detoxification, glutathione synthesis, and defense against oxidative damage. Stress-activated MAP kinases p38, ERK, and JNK cooperate in the efficient nuclear accumulation of Nrf2 in a cell-type-dependent manner. Activation of p38 induces membrane trafficking of a glutathione sensor neutral sphingomyelinase 2, which generates ceramide upon depletion of cellular glutathione. We previously proposed that caveolin-1 in lipid rafts provides a signaling hub for the phosphorylation of Nrf2 by ceramide-activated PKCζ and casein kinase 2 to stabilize Nrf2 and mask a nuclear export signal. We further propose a mechanism of facilitated Nrf2 nuclear translocation by ERK and JNK. ERK and JNK phosphorylation of Nrf2 induces the association of prolyl cis/trans isomerase Pin1, which specifically recognizes phosphorylated serine or threonine immediately preceding a proline residue. Pin1-induced structural changes allow importin-α5 to associate with Nrf2. Pin1 is a co-chaperone of Hsp90α and mediates the association of the Nrf2-Pin1-Hsp90α complex with the dynein motor complex, which is involved in transporting the signaling complex to the nucleus along microtubules. In addition to ERK and JNK, cyclin-dependent kinase 5 could phosphorylate Nrf2 and mediate the transport of Nrf2 to the nucleus via the Pin1-Hsp90α system. Some other ERK target proteins, such as pyruvate kinase M2 and hypoxia-inducible transcription factor-1, are also transported to the nucleus via the Pin1-Hsp90α system to modulate gene expression and energy metabolism. Notably, as malignant tumors often express enhanced Pin1-Hsp90α signaling pathways, this provides a potential therapeutic target for tumors.

## 1. Introduction

Cells respond to mild oxidative stress, such as non-lethal levels of hydrogen peroxide (H_2_O_2_), and rapidly upregulate defense systems to afford protection against oxidative damage (reviewed in [1]). Among the redox-sensitive transcription factors, nuclear factor-E2-related factor 2 (Nrf2) plays an important role in the expression of genes involved in detoxification, glutathione (GSH) synthesis, and defenses against oxidative damage [2,3,4]. Upregulation and activation of Nrf2 are controlled through a complex transcriptional, translational, and post-translational network that ensures an increase in its activity during redox perturbation, inflammation, growth factor stimulation, and nutrient/energy fluxes, thereby enabling Nrf2 to orchestrate adaptive responses to diverse forms of stress (reviewed in [5,6,7,8,9]).

The rapid activation of Nrf2 by stress agents depends on both inhibition of degradation and/or enhanced Nrf2 translation in the cytoplasm and facilitated nuclear translocation of Nrf2 to the nucleus (Figure 1A). Nrf2 is unstable under unstressed conditions due to constant degradation via the 26S proteasome, which is mediated by its cytoplasmic binding partner Kelch-like ECH-associated protein 1 (Keap1). Keap1 has cysteine residues highly reactive with various types of electrophiles, which form Michael adducts with Keap1 to inhibit interaction with Nrf2 resulting in Nrf2 stabilization and accumulation (reviewed in [10,11,12]). Degradation of Nrf2 is also controlled by phosphorylation at the Neh6 domain by glycogen synthase kinase-3 (GSK-3) (reviewed in [6,7,8]). It enhances the degradation of Nrf2 mediated by β-transducin repeat-containing protein, present in the ubiquitin ligase complex. Therefore, inhibition of GSK-3 by PI3-K (phosphatidylinositol 3-kinase)/Akt (PKB) signaling is a prerequisite to induce Keap1-mediated Nrf2 stabilization by electrophiles (reviewed in [6,7,8]).

In addition to Nrf2 stabilization, oxidative stress induces rapid Nrf2 protein synthesis [13]. Treatment of rat cardiomyocytes, human HeLa, and other cells with H_2_O_2_ induces rapid upregulation of translational Nrf2 protein synthesis independent of Nrf2 protein stabilization in these cells [13,14]. Notably, Nrf2 mRNA has an internal ribosomal entry site within the 5′ untranslated region (5′UTR) of human Nrf2 mRNA [14], and H_2_O_2_ treatments have been shown to cause rapid translocation of La autoantigen (Sjögren Syndrome Antigen B) from the nucleus to perinuclear space to associate with ribosomes. Binding La autoantigen to 5′UTR of Nrf2 mRNA stabilizes it and induces an Internal Ribosome Entry Site mediated protein translation in HeLa cells [15]. However, increased Nrf2 levels in the cytoplasm do not automatically lead to nuclear accumulation of Nrf2, as the transport of Nrf2 into the nucleus is tightly controlled, requiring additional regulators such as importins [16].

Nuclear import of proteins is tightly controlled by the interaction with a cytosolic protein termed importin composed of α and β subunits [17,18,19]. Importin α recognizes and binds nuclear localization signals (NLSs) of karyophilic proteins, and importin β helps bind them to the nuclear envelope, with energy-dependent, small GTPase Ran-mediated translocation through the pore resulting in the accumulation of import substrate and importin-α in the nucleus [18,19,20]. Nrf2 has three functional Lys/Arg-rich NLS motifs in Neh1, Neh2, and Neh3 domains, respectively, and mutations of the three NLS motifs significantly impair nuclear translocation of Nrf2 in HepG2 cells, and ARE-reporter gene expression in human leukemia K562 cells treated with *tertiary*-butylhydroquinone (tBHQ) [16]. These authors further showed that anti-importin α5 and anti-importin β1, respectively, co-immunoprecipitated with Nrf2 shortly after treating K562 cells with tBHQ, and that the amount of Nrf2 co-immunoprecipitated by these antibodies in nuclear fractions increased over 30–60 min [16]. This study shows that the transcriptional activation of Nrf2 depends largely on importin-mediated nucleocytoplasmic shuttling. It is also noted that exportin mediates nuclear export through interaction with nuclear export signals (NESs) [21,22]. Nrf2 has at least two functional NESs in the leucine zipper domain [23] and the transactivation domain [24]. The former Nrf2 NES can be masked by the formation of heterodimers with functional partner small Maf proteins, and the latter is a redox-sensitive NES. Thus, maximal Nrf2 activation or nuclear accumulation is achieved by enhanced association with importin α5β1 in the cytoplasm to facilitate nuclear translocation and inhibition of exportin binding in the nucleus (Figure 1B).

Although the precise mechanism of importin-dependent Nrf2 nuclear translocation remains unclear, previous studies have established the importance of stress-activated mitogen-activated protein kinases (MAPKs), p38, extracellular signal-regulated kinase (ERK), and c-Jun N-terminal kinase (JNK) in the nuclear translocation and activation of Nrf2 (reviewed in [25,26]). MAPKs are proline-directed serine/threonine protein kinases activated by dual phosphorylation on threonine and tyrosine residues in response to a wide array of extracellular stimuli. They are essential components of signaling pathways that convert various extracellular signals into intracellular responses through serial phosphorylation cascades (reviewed in [27,28,29]). ERK, known alternatively as microtubule-associated protein-2/myelin basic protein kinase, is activated by numerous hormones, growth factors, and other extracellular stimuli (reviewed in [30,31]), whereas JNK and p38 are activated by distinct and overlapping sets of stress-related stimuli, including heat shock, inflammatory cytokines, ultraviolet, gamma irradiation, and hyperosmolarity (reviewed in [32]). Interestingly, tBHQ activates ERK2 and JNK1 in HepG2 and HeLa cells [33], and phenethyl isothiocyanate activates ERK2 and JNK1, which phosphorylate Nrf2, resulting in nuclear translocation in human prostate cancer PC-3 cells [34] (Figure 1C). Table 1 summarizes the role of MAPKs in Nrf2 activation and ARE-mediated reporter gene expression in different cell types stimulated with various stress agents reported between 2000 to 2012 [34,35,36,37,38,39,40,41,42,43,44,45,46,47,48,49,50,51,52,53,54]. Evidently, the dependence of Nrf2 activation on ERK, JNK, and p38 MAPKs differs significantly among cells and stress agents (see Table 1). The dependence of Nrf2 activation on MAP kinases is widely confirmed by later numerous studies using different types of cultured cells stimulated with other various natural and synthetic chemical agents that activate Nrf2. However, molecular mechanisms of MAP kinase-dependent Nrf2 activation remain unsolved.

In this review, we critically evaluate the differential role of p38, JNK, and ERK on the activation of Nrf2 by stress agents. We hypothesize that ERK/JNK signaling contributes to the nuclear translocation of Nrf2 and that simultaneous activation of p38 signaling activated under glutathione (GSH) deletion leads to maximal accumulation of Nrf2 in the nucleus (Figure 1D). We previously proposed that p38 signaling leads to the phosphorylation of Nrf2 by PKCζ and casein kinase 2 (CK2) to stabilize it and masks a nuclear exporting signal (NES) [1] as described in Section 2. Notably, previous studies show that stress-activated ERK/JNK signaling facilitates Nrf2 nuclear translocation [34,48,50]. We discuss the possible functional partners of ERK that regulate facilitated Nrf2 nuclear translocation, and propose a novel mechanism of Nrf2 nuclear translocation mediated by ERK, involving peptidylprolyl cis/trans isomerase (PPIase) NIMA-interacting 1 (Pin1) and the molecular chaperon heat shock protein 90α (Hsp90α) as discussed in Section 3, Section 4, Section 5 and Section 6. We further discuss cyclin-dependent kinase 5 (Cdk5)-mediated Nrf2 nuclear translocation in Section 7.

## 2. p38 Controls Glutathione Sensor Neutral Sphingomyelinase 2

Maintenance of the small antioxidant GSH at high levels is essential for protecting cells against oxidative damage. Cells respond to the downregulation of GSH under oxidative stress by activating Nrf2, which upregulates the expression of genes required for GSH synthesis to restore cellular GSH levels. Notably, neutral sphingomyelinase 2 (nSMase2) senses GSH levels, and its activity is inhibited by high levels of cellular GSH (>3mM), but depletion of cellular GSH induces nSMase2 activation to generate the lipid signaling molecule ceramide [55,56,57,58,59]. We previously proposed that depletion of GSH induces ceramide/PKCζ/CK2 signaling leading to phosphorylation of Nrf2 by these kinases [60,61,62]. Interestingly, oxidative stress induces activation of p38 MAP kinase, which causes trafficking of nSMase2 from perinuclear regions to the plasma membrane [63,64], thereby enhancing ceramide generation by nSMase2 under oxidative stress. We further speculated that ceramide/PKCζ/CK2 signaling phosphorylates Nrf2 tethered to caveolin 1 (Cav1) in membrane lipid rafts/caveolae and that phosphorylation by these kinases stabilizes Nrf2 and masks an NES which could favor Nrf2 nuclear localization [1] (Figure 2A).

However, this hypothesis requires an additional mechanism to facilitate the translocation of stabilized Nrf2 from cell membrane compartments to the nucleus. Facilitated nuclear translocation of Nrf2 requires importin binding to the NLSs and directional movement through the cytoplasm to nuclear membrane pores. Upon arrival at the nuclear pores, the complexes are transferred to the nuclear interior by importin-dependent facilitated diffusion [20]. Mechanisms and functional partners for the facilitated movement of Nrf2 from the cell periphery to nuclear pores remain unclear. We discuss the possible functional partners and mechanism of facilitated Nrf2 nuclear translocation in the following sections.

## 3. ERK/JNK and PPIase Pin1 Control Nrf2 Nuclear Translocation

Xu et al. [34] previously showed that ERK and JNK modulate the nuclear translocation of Nrf2 in human prostate cancer PC-3 cells. These authors pre-treated the cells in 0.5% serum-containing medium overnight, which induced sequestration of Nrf2 in the cytoplasm. Upon transfection of JNK1 and its activating kinase MKK into PC-3 cells, Nrf2 was localized in both the cytoplasm and nucleus. Similar results were obtained when PC-3 cells were transfected with ERK2 and its activating kinase MEK1. These authors suggested that JNK1 and ERK2 can phosphorylate Nrf2 and induce nuclear translocation.

Recent studies show that the PPIase Pin1 upregulates nuclear accumulation of Nrf2 [65,66,67]. PPIase catalyzes the conversion between cis and trans conformations of proline imidic peptide bonds, playing a role in protein folding, signal transduction, trafficking, assembly, and cell cycle regulation [68]. The three classes of PPIase are cyclophilins, FK506-binding proteins (FKBPs), and parvulins [68], and Pin1 belongs to the parvulin class of PPIase. Cyclophilins and FKBPs are called immunophilins as immunosuppressive drugs such as cyclosporin A, FK506, and rapamycin directly bind and inhibit these PPIases [68]. Previous studies show that immunophilins associate with steroid hormone receptors to modulate their functions (reviewed in [69]).

Pin1 is a unique PPIase that specifically recognizes phosphorylated serine or threonine immediately preceding a proline residue (pSer/Thr-Pro), isomerizes the peptide bond, and is known to play an important role in cell cycle progression (reviewed in [68,70]). Liang et al. [65] showed that Pin1 contributes to Nrf2 activation in pancreatic ductal adenocarcinoma cells with high K-ras activity [65]. Saeidi et al. [66] found that enhanced H-ras signaling in human breast cancer cells induces the association of Pin1 with Nrf2 to protect Nrf2 from Keap1-mediated degradation. Saeidi et al. [66] further showed that phosphorylation of human Nrf2 at Ser-215, -408, and -577 is essential for its interaction with Pin1 [66]. These authors showed that among MAP kinases, ERK and JNK, but not p38, phosphorylate these serine residues, suggesting Ras-ERK signaling promotes Pin1 association with ERK-phosphorylated Nrf2 to facilitate translocation to the nucleus. We speculated, based on these studies, that ERK2 and JNK initially phosphorylate Nrf2, resulting in the association of Pin1 to the p-Ser/Thr-Pro site(s). It seems plausible that a Pin1-mediated structural change in Nrf2 could expose NLS(s) for association with importin α5, which then recruits importin β1 to facilitate nuclear translocation (Figure 2B). We next discuss the possible additional partners for the ERK/JNK-Pin1-mediated Nrf2 nuclear translocation.

## 4. PPIase and Hsp90 Cooperate in the Nuclear Transport of Signaling Molecules

Another important function of some PPIases is their interaction with heat shock protein 90 (Hsp90) and the dynein/dynactin complex [71,72]. Hsp90 is the major molecular chaperone protecting many client proteins from denaturation and aggregation (reviewed in [73,74,75]). Notably, in addition to chaperone activity, Hsp90 controls nucleocytoplasmic trafficking of signaling molecules (reviewed in [76,77]). PPIases are regarded as co-chaperones of Hsp90 and are crucial for translocating hormone receptors, transcription factors, and signaling molecules [77,78,79,80,81].

Hsp90 has three functional domains, an N-terminal domain with ATPase, a middle domain for binding co-chaperones and clients, and a C-terminal domain for dimerization [82] (Figure 3A). The Hsp90 dimer forms two alternative structures, ATP-free open and ATP-bound closed structures [83,84] (Figure 3A). Notably, ATP hydrolysis is coupled with the chaperone activity and accompanies the release of associated clients. There are many co-chaperone proteins (cofactors) that regulate Hsp90 functions, including its ATPase activity [85]. Importantly, to keep a client associated with the benefits of long-distance translocation, a co-chaperone p23 inhibits ATPase activity [86] and is recruited to the Hsp90-FKBP52 complex [81,87].

Two isoforms of Hsp90, α and β, are abundantly expressed in the cytoplasm with similar chaperone activities. Proliferating cells express higher levels of Hsp90α than Hsp90β, as Hsp90α gene expression is controlled by c-Myc downstream of growth factor signaling [88]. As Hsp90α has higher dimer-forming potential than Hsp90β [89,90], and p23 co-chaperone preferentially associates with Hsp90α compared to Hsp90β [84], we propose that the Hsp90α dimer and p23 co-chaperone compose a backbone for the nuclear transport machinery for co-transport of PPIase and client proteins (Figure 3B).

## 5. Functional Interaction of Nrf2 with Hsp90

Ngo et al. [91] recently showed that Hsp90 (isoforms not identified) directly interacts with Nrf2 in a yeast model expression system and HeLa cells, and with Hsp90 preventing overexpressed Nrf2 from forming protein aggregates. This study shows that Nrf2 is a client of Hsp90 and suggests a possibility that under the normal/low levels of Nrf2, the Pin1-Nrf2 associates with Hsp90α dimer-p23, forming a multiprotein complex that can be transported efficiently to the nuclear pore complex via the dynein motor complex along microtubules (Figure 3C). To verify this hypothesis, further studies are required to detect stable interaction of the Nrf2-Pin1 complex with Hsp90α and to demonstrate the Pin1-mediated linkage of the signaling complex with the dynein motor system.

Interestingly, previous studies suggest functional interactions of Nrf2 with Hsp90. Jia et al. [92] observed that an Hsp90 ATPase inhibitor, 17-dimethylaminoethylamino-17-demethoxygeldanamycin (17-DMAG), upregulated nuclear Nrf2 and the expression of HO-1 in neuronal HT22 cells subjected to hypoxia/reoxygenation. Lazaro et al. [93] also observed that 17-DMAG upregulated Nrf2 activation in macrophages and vascular smooth muscle cells in atherosclerotic plaques in diabetic apolipoprotein E-deficient mice. In contrast to these reports, Hsp90 ATPase inhibitors, including 17-allylamino-demethoxygeldanamycin (17-AAG), caused the gradual death of human cancer cells expressing high levels of Nrf2 [94]. The Hsp90 inhibitor exhibits no toxicity to other cancer cells with normal levels of Nrf2 but induces toxicity in cells after treatment with diethyl maleate (100 μM) to upregulate Nrf2 levels [94]. This toxic effect of 17-AAG may reflect the importance of Hsp90 chaperone activity to prevent Nrf2 protein aggregation in Nrf2-overexpressing cells [91].

## 6. Pin1 Controls the Nuclear Translocation of Other ERK Substrates

In addition to Nrf2, the ERK-Pin1 system naturally controls the nuclear translocation of several other ERK substrates, which are important for energy metabolism and proliferation. The nuclear translocation of pyruvate kinase M2 (PKM2) is controlled by the ERK-Pin1 system [95]. PKM2 promotes glucose metabolism by aerobic glycolysis and contributes to anabolic metabolism [96]. PKM2 expression is upregulated in multiple cancer types and contributes to the Warburg effect (reviewed in [97,98,99,100,101]). Yang et al. [102] showed that activated ERK2 binds directly to PKM2 through the ERK2 docking groove and phosphorylates PKM2 at Ser-37, inducing a structural change of PKM2 from tetramer to dimer or monomer. PKM2 dimer recruits Pin1 for cis-trans isomerization of PKM2 and binding to importin α5 and translocation to the Hsp90 containing complex to the nucleus [102], supporting our hypothesis of ERK-Pin1-Hsp90 dependent Nrf2 nuclear translocation (Figure 3C).

The ERK-Pin1-Hsp90α machinery also controls the nuclear transport of hypoxia-inducible transcription factor-1 (HIF-1α). HIF-1α mediates the activation of networks of target genes involved in angiogenesis, erythropoiesis, and glycolysis (reviewed in [103,104,105]). Besides hypoxic conditions, phosphorylation of HIF-1α at Ser-451 by ERK is another central post-translational modification, which regulates its stability under both hypoxia and physiological normoxia (reviewed in [106]), and plays a crucial role in promoting tumor growth [107,108,109]. Jalouli et al. [108] showed that Pin1 associates with the p-Ser-Pro motif and regulates HIF-1α transcriptional activity. HIF-1α is a client of Hsp90α [110] and contains NLSs for importin α binding [111,112,113]. Mylonis et al. [114] further showed that phosphorylation of Ser-641/643 by ERK inhibits the association of exportin to the NES, resulting in the accumulation of HIF-1α in the nucleus. These studies indicate that the nuclear translocation of HIF-1α is largely dependent on the ERK-mediated Pin1-Hsp90α system.

## 7. Cdk5 Controls Nrf2 Nuclear Translocation through Pin1

In addition to ERK and JNK, another proline-directed serine/threonine kinase, cyclin-dependent kinase 5 (Cdk5), also controls Nrf2 activation. Jimenez-Blasco et al. [115] showed that treating astrocytes from fetal rat brains with 20 μM N-methyl-d-aspartate (NMDA) for 8 h induced Nrf2-dependent activation of antioxidant genes. These authors further showed that NMDA induces the phospholipase C-mediated endoplasmic reticulum release of Ca^2+^ and activation of PKCδ, which phosphorylates and stabilizes Cdk5 cofactor p35. Furthermore, the active p35/Cdk5 complex phosphorylated Nrf2 leading to Nrf2 nuclear translocation and ARE-mediated gene expression [115]. In another study, Lee et al. [116] showed that tBHQ induces Nrf2 activation and expression of NQO1 in IMR-32 human neuroblastoma cells independent of ERK. Interestingly, oxidative stress induces the upregulation of Cdk5 catalytic subunit p35 in IMR-32 cells [117], suggesting Cdk5/p35 instead of ERK plays a role in Pin1-mediated Nrf2 nuclear translocation in neuroblastoma cells. It has been reported that Cdk5 phosphorylates ubiquitin ligase TRIM59 leading to Pin1 and importin α5 association resulting in nuclear translocation [118]. These results suggest that Cdk5/Nrf2/Pin1 axis contributes to Nrf2 nuclear translocation, like ERK/Nrf2/Pin1 axis, irrespective of whether Cdk5 phosphorylates different Ser/Thr-Pro sites of Nrf2. Cdk5 regulates neuronal functions but is also associated with cancer development and has been considered a potential target for cancer treatment (reviewed in [119,120,121,122,123]).

## 8. Summary and Conclusions

MAP kinases contribute to the activation of the transcription factor Nrf2 (reviewed by Kong et al., 2001, 2002), noting that activation is dependent on the cell type and stress agent (see Table 1). We proposed that p38-mediated signaling could stabilize Nrf2 via ceramide-activated PKCζ phosphorylation and mask an NES by CK2 phosphorylation under oxidative stress accompanying GSH depletion (Figure 2A) [1]. In contrast, ERK2 and JNK1 induce Nrf2 translocation into the nucleus [34]. Concerning the mechanism underlying ERK/JNK-dependent Nrf2 nuclear translocation, we propose a novel concept that direct phosphorylation of Nrf2 by ERK/JNK induces assembly of the Hsp90α-Pin1-Nrf2 complex. Pin1 causes prolyl-isomerization of Nrf2, which allows importin α5 to associate the Hsp90α-Pin1-Nrf2 complex. Then, the Hsp90α-Pin1-Nrf2-importin containing signaling complex is carried by the dynein motor system toward the nuclear pore complex along microtubules (Figure 3C). In addition to ERK and JNK, Cdk5 also phosphorylates Nrf2 and helps Nrf2 nuclear translocation via Hsp90α-Pin1-dynein machinery. Importantly, Hsp90 contributes to Nrf2 activation in two ways, inhibition of denaturation or aggregation when Nrf2 is over-expressed [91] and facilitation of Nrf2 nuclear import. The Hsp90 chaperone activity depends on ATP hydrolysis, but it accompanies client release. However, ATP hydrolysis is suppressed during nuclear transport of Hsp90α-Pin1-Nrf2 containing multiprotein complex via association of co-chaperon p23 (Figure 3C). Thus, under oxidative stress, p38 and ERK/JNK and/or Cdk5 could work together to stabilize Nrf2 to facilitate the maximal accumulation of Nrf2 in the nucleus (Figure 1C,D).

Hsp90α is required for the proliferation, migration, and invasion of cancer cells in culture [124]. Hsp90 is considered a druggable target for cancer treatment [125,126]. Accumulating evidence indicates that Pin1 plays a key role in various cancers. Pin1-mediated β-catenin accumulation occurs in about 70% of hepatocellular carcinoma [127]. Intriguingly, cell proliferation, migration, and invasion are significantly inhibited in Pin1-silenced Hep-2 cells [128]. Silencing of Pin1 causes down-regulation of β-catenin and cyclin D1 expression [128] and significantly increases the sensitivity to cisplatin in HeLa cells [129]. Thus, ERK- and Cdk5-signaling coupled with Hsp90α-Pin1-dynein machinery may be a prime target of chemotherapy for tumors. Developing chemical agents that inhibit Pin1 rather than Hap90 ATPase activity could be a promising approach for cancer chemotherapy (reviewed in [130,131,132]). For instance, a Pin1 inhibitor all-trans retinoic acid has been used to treat acute promyelocytic leukemia in animal models and human patients [133], and all-trans retinoic acid reduced the growth of transplanted tamoxifen-resistant human breast cancer cells in mice [134]. Dubiella et al. [135] showed that a Pin1 inhibitor Sulfopin reduced tumor progression and conferred survival benefits in animal models, while Liu et al. [136] developed a delivery system of a Pin1 inhibitor AG177724 targeting cancer-associated fibroblasts and observed the inhibition of tumor growth in mice.

Kim et al. [137] raised a question concerning the role of Pin1 in the expression of HO-1 induced by nitric oxide (NO) in mouse vascular smooth muscle cells and embryonic fibroblasts. These authors observed that treatment of the cells with NO donor nitroprusside (3 mM) for 8 h upregulated HO-1 levels in control cells to a higher extent than in Pin1 deficient cells and argued that Nrf2/ARE-mediated transcriptional activity is negatively controlled by Pin1 in fibroblasts [137]. However, the expression of HO-1 is controlled by both Nrf2 and AP-1 via partially overlapping AREs and TPA-responsive elements in the gene promotor [138], and Mouawad et al. [139] showed that NO-dependent expression of HO-1 is controlled by transcription factors C/EBPβ and AP-1 in macrophages [139]. Therefore, we suggest the possibility that NO-activated AP-1 could induce HO-1 gene expression more efficiently in the absence of Pin1-mediated Nrf2 nuclear translocation.

Low molecular kinase inhibitors are widely used to examine the function of kinases, but some inhibitors exhibit unexpected side effects. For instance, the p38 inhibitor SB203580 is a potent agonist of aryl hydrocarbon receptor (AhR) [140,141], which is a ligand-activated transcription factor and key regulator of xenobiotic metabolism, and AhR activation induces Cyp1a1 gene expression via xenobiotic responsive element (XRE) in Hepa 1c1c7 and HepG2 cells [142]. A complex effect of SB203580 is observed in HepG2 cells, which express high AhR levels [143]. Yu et al. [143] observed that tBHQ (100 μM) activated p38 and upregulated NQO1 expression in HepG2 cells, suggesting the role of the p38/Nrf2/ARE axis for the NQO1 expression. However, these authors observed that the addition of the p38 inhibitor SB203580 (5 μM) with tBHQ upregulated expression of NQO1 levels higher than tBHQ alone in 24 h, and argued that the p38 kinase pathway functions as a negative regulator in the ARE-mediated induction of phase II detoxifying enzymes [143]. We believe this conclusion may be misleading due to neglecting the influence of AhR activation by SB203580. Notably, NQO1 gene expression can be controlled by both Nrf2/ARE and AhR/XRE [144], and Nrf2 gene expression can be upregulated by AhR/XRE signaling [145]. Treatment of hepatoma 1c1c7 cells with AhR agonist 2,3,7,8-tetrachlorodibenzo-*p*-dioxin (10 nM) leads to upregulation of Nrf2 mRNA in 2 h and Nrf2 protein levels in 6 h [145]. Notably, tBHQ is a ligand for AhR. It is known to induce expression of Cyp1a1 in hepatoma 1c1c7 and HepG2 cells [146], suggesting the possibility that SB203580 and tBHQ synergistically activate AhR/XRE signaling, leading to rapid NQO1 gene expression with delayed Nrf2/ARE-mediated NQO1 expression. Thus, Phase I and II xenobiotic metabolism is coordinately regulated by the cross-talk between AhR and Nrf2 via ARE and XRE elements, respectively (reviewed in [147]).

In summary, we propose that ERK, JNK, and Cdk5 control Nrf2 phosphorylation inducing a formation of a multiprotein complex containing Hsp90α-p23-Pin1-Nrf2-importins to associate with dynein motors to move from cell membrane signaling compartments toward nuclear pores along microtubules (Figure 3C). Therefore, the integrity of microtubules is important to ensure the nuclear translocation of Nrf2 and other signaling molecules. It is important to note that tau family proteins control the assembly and maintenance of the structural stability of microtubules and that proline-directed kinases, MAP kinases, GSK-3β, and Cdk5, can phosphorylate tau proteins and affect their functions (reviewed in [148,149,150]). GSK-3β phosphorylated tau reduces the affinity to microtubules, causing the dysfunction of microtubules [151,152]. Interestingly, Pin1 interacts with phosphorylated tau proteins, restores the ability of tau to bind microtubules and promote assembly in vitro [153], and downregulates the GSK-3β-mediated phosphorylation of tau [154,155]. These studies suggest that Pin1/tau axis is also important for microtubule-mediated facilitated translocation of Nrf2 to the nucleus. As tau controls axonal transport and neurite outgrowth in neurons, defects in tau function could lead to neurodegeneration [156]. Thus, Pin1 plays an important but opposite role in the pathogenesis of Alzheimer’s disease and many human cancers (reviewed in [157,158]).

## Figures and Tables

**Figure 1 antioxidants-12-00274-f001:**
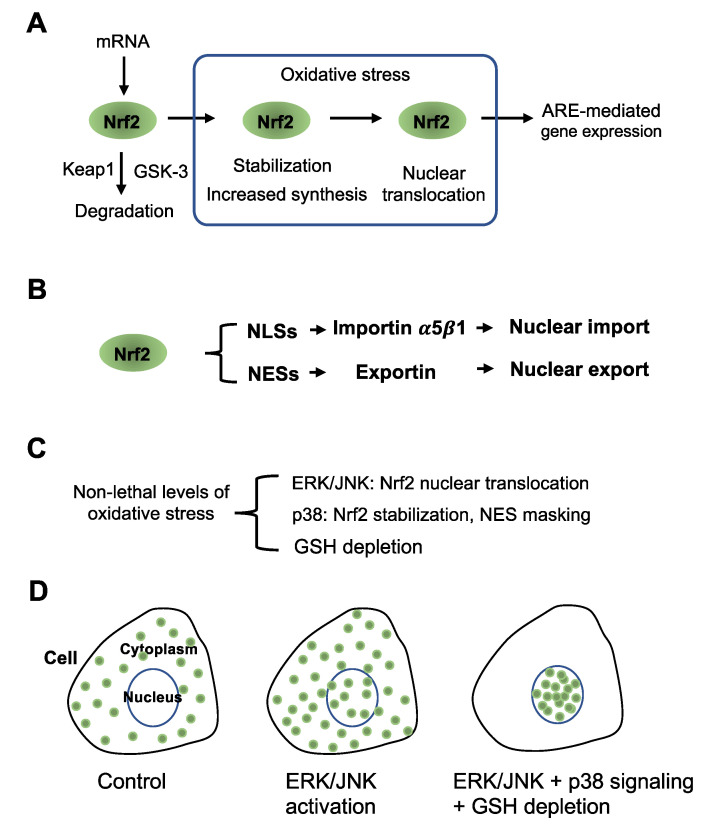
**MAP kinases p38, ERK, JNK, and GSH depletion control activation of the transcription factor Nrf2.** (**A**) Rapid activation of Nrf2 under oxidative stress depends on two separate steps, stabilization and upregulation of Nrf2 protein and nuclear translocation. (**B**) Nuclear import of Nrf2 is controlled by the association of importin α5β1 to its nuclear localization signals (NLSs), while nuclear export of Nrf2 is controlled by exportin binding to its nuclear export signals (NESs). (**C**) Non-lethal levels of oxidative stress, such as H_2_O_2,_ induce GSH depletion and activation of MAP kinases (reviewed in [1]). ERK and JNK control Nrf2 nuclear translocation, and p38 signaling induces stabilization of Nrf2 and masking of an NES. (**D**) In unstressed conditions, Nrf2 (green dots) levels are low and mainly localized in the cytoplasm but not in the nucleus due to functional nuclear export signals. ERK/JNK activation could cause nuclear transport of Nrf2 across the nuclear membrane but does not enable Nrf2 to accumulate in the nucleus due to the functional nuclear export signals. As p38 signaling masks a nuclear export signal, simultaneous activation of ERK and p38 signaling pathways under GSH-depleting conditions induces effective nuclear accumulation/activation of Nrf2.

**Figure 2 antioxidants-12-00274-f002:**
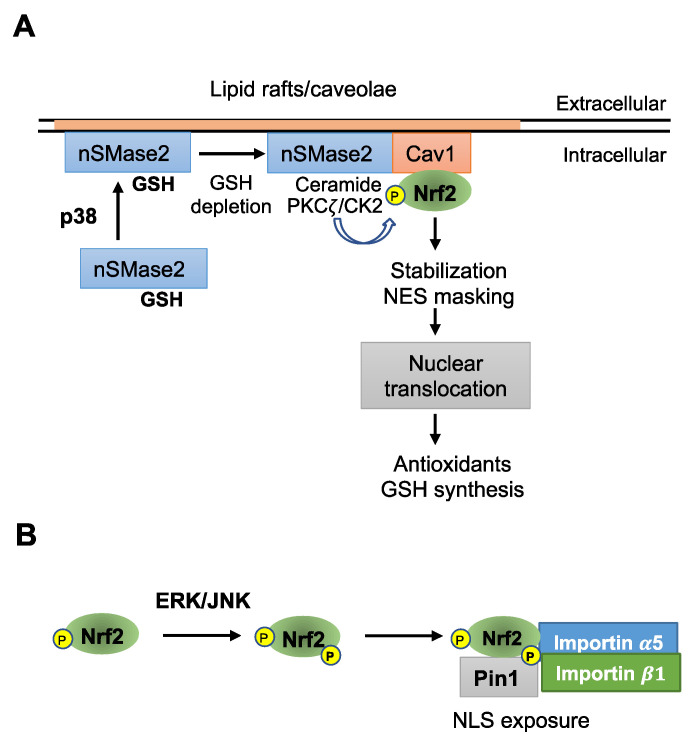
**Stress-activated p38 controls nSMase2 activation, and ERK and JNK mediate Pin1-dependent Nrf2 nuclear translocation**. (**A**) MAP kinase p38 induces the transfer of neutral sphingomyelinase 2 (nSMase2) from the perinuclear region to the plasma membrane. Glutathione (GSH) depletion causes ceramide/PKCζ/CK2 signaling, which induces Nrf2 phosphorylation and stabilization and masks a nuclear export signal [1]. (**B**) Phosphorylation of Nrf2 by ERK/JNK leads to an association with Pin1, which causes a cis/trans structural change at the preceding proline residue (pSer/Thr-Pro), allowing an association with importin α5 at the exposed nuclear localization signal. Then, importin β1 associates with importin α5, allowing Nrf2 to translocate to the nucleus.

**Figure 3 antioxidants-12-00274-f003:**
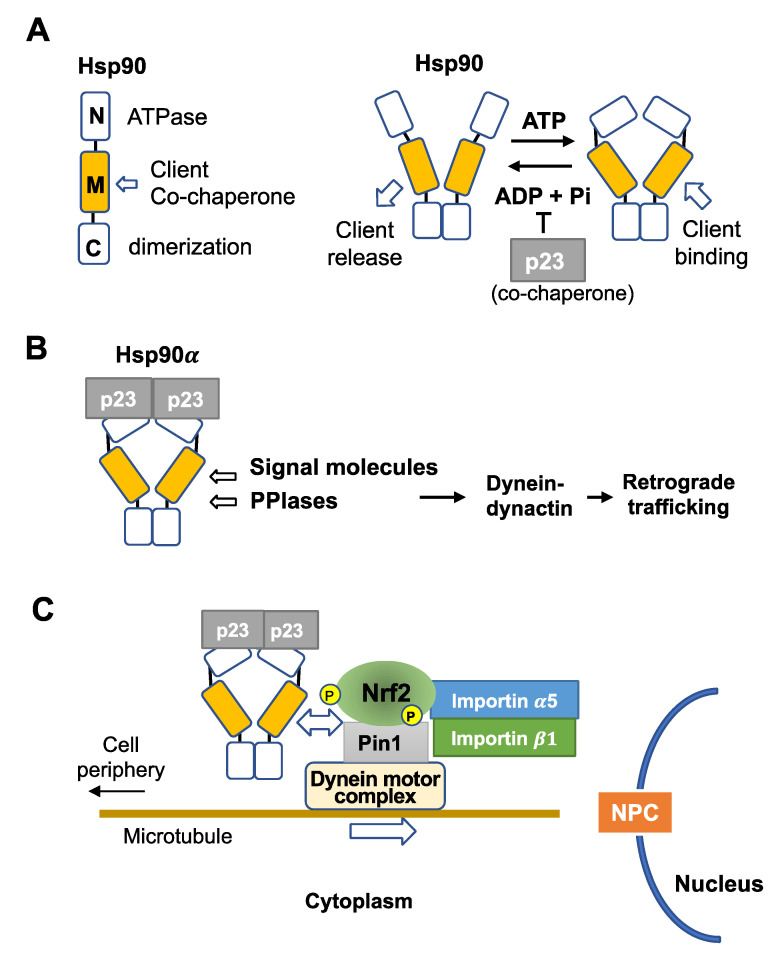
**Proposed mechanism of Hsp90-mediated nuclear translocation of Nrf2.** (**A**) Hsp90 is composed of three domains, an N-terminal domain with ATPase activity, a middle domain for binding co-chaperones and clients, and a C-terminal domain for dimerization. Hsp90 dimer takes two forms, e.g., open and ATP-bound closed ring forms. Co-chaperone p23 inhibits Hsp90 ATPase activity. (**B**) Hsp90α dimer-p23 complex is the basal structure for transporting PPIase-associated signal molecules such as steroid receptors to the nucleus. (**C**) Hsp90α-p23-Pin1-Nrf2-importin-α5β1 containing complex associates with a dynein motor complex, which carries the cargo toward the nuclear pore complex (NPC) along microtubules.

**Table 1 antioxidants-12-00274-t001:** MAP kinase-dependent Nrf2 activation.

Cells	Activators	MAPK Dependence	References
Human hepatoma HepG2	Sodium arsenite and mercury chloride	JNK-dependent ARE reporter gene and HO-1 expression	[35]
	Pyrrolidine dithiocarbamate	ERK and p38 inhibitors PD98059 and SB202190 reduced about 50% in γ-glutamylcystein synthetase expression	[36]
	Diallyl sulfide	ERK- and p38- dependent Nrf2 nuclear translocation and HO-1 expression	[37]
	Gallic acid	p38 inhibitor reduced ARE-dependent P-form of phenol sulfotransferase expression	[38]
Human hepatocyte	Quercetin	p38- and ERK-dependent Nrf2 activation and HO-1 expression	[39]
Human mammary epithelia MCF-7	Cadmium chloride	p38-dependent but ERK-independent HO-1 expression	[40]
Human HeLa	Phenethyl isothiocyanate	JNK-dependent ARE-reporter gene expression	[41]
Human monocytic THP-1	α-Lipoic acid	p38 inhibitor significantly reduced Nrf2 dependent HO-1 expression	[42]
Human aortic smooth muscle	Oxidized low-density lipoprotein	ERK, p38, and JNK inhibitors respectively reduced HO-1 expression and Nrf2 nuclear translocation	[43]
Human prostate carcinoma PC-3	Phenethyl isothiocyanate	ERK and JNK phosphorylate Nrf2 and induce nuclear translocation of Nrf2	[34]
Human monocyte	Curcumin	p38 inhibitor but not ERK inhibitor reduced ARE-dependent GCLM and HO-1 mRNA expression	[44]
Mouse alveolar epithelial C10	Hyperoxia	Hyperoxia activates NADPH oxidase, which results in ERK-dependent Nrf2 activation	[45]
Mouse macrophage RAW 264.7	Lipopolysaccharide	p38 inhibitor significantly reduced Nrf2 dependent HO-1 expression	[46]
Mouse cochlear	Piperine	JNK inhibitor significantly reduced ARE-reporter gene expression and HO-1 expression	[47]
Mouse keratinocyte	3H-1,2-dithiole-3-thione	ERK inhibitor but not p38 inhibitor suppressed Nrf2 nuclear translocation and ARE-reporter gene expression	[48]
Rat epithelial L2	4-hydroxynonenal	ERK- and p38-dependent EPRE-mediated γ-glutamyl transpeptidase expression	[49]
Rat vascular smooth muscle	15d-PGJ_2_	p38 inhibitor abolished Nrf2 dependent HO-1 expression	[50]
Rat primary hepatocytes	Methionine restriction	ERK-dependent Nrf2 nuclear translocation and GSH-*S*-transferase π expression	[51]
Rat kidney epithelial NRK-52E	Curcumin	p38 inhibitor reduced about 50% of HO-1 expression, but ERK and JNK inhibitors did not suppress HO-1 expression	[52]
Bovine aortic endothelial	Spermine NONOate (NO donor)	p38 and ERK inhibitors SB203580 and PD98059 respectively reduce HO-1 expression	[53]
Guinea pig gastric mucosal	Indomethacin	p38 inhibitor significantly reduced Nrf2 nuclear accumulation and HO-1 expression	[54]

Abbreviations: ARE, antioxidant response element; HO-1, heme oxygenase-1; GCLM, glutamate-cysteine ligase modifier subunit; EPRE, electrophile response element.

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
