# Peer review of "Stress Activated MAP Kinases and Cyclin-Dependent Kinase 5 Mediate Nuclear Translocation of Nrf2 via Hsp90α-Pin1-Dynein Motor Transport Machinery"

_antioxidants, 2023, doi:10.3390/antiox12020274_

Round 1

Reviewer 1 Report

The article entitled “Stress activated MAP kinases and cyclin-dependent kinase 5 mediate nuclear translocation of Nrf2 via Hsp90α-Pin1-dynein motor transport machinery” is a very complete and focused review on the regulation of the transcription factor at cellular level which is based on its post-translational modification and interaction with proteins.

It is a well written article and in my opinion it is fully eligible for publication in the present form.

Author Response

Thank you for your positive comments to our manuscript. I have corrected minor errors. P5, do (does), co-immunoprecipitated (co-immunoprecipitate), P16; via (vis), p17, deleted ")" after cells.

Reviewer 2 Report

The review presented by Ishii and colleagues entitled “Stress activated MAP kinases and cyclin-dependent kinase 5 mediate nuclear translocation of Nrf2 via Hsp90alfa-Pin1-dynein motor transport machinery” affords a complete summarize about the nuclear translocation of Nrf2 with current studies performed in different cell types, above all focused on cancer cell lines. The review is suitable for publication as soon as few considerations should be taken into account.

Major points:

-  Authors focus on the role in cancer of this transcription factor. Nonetheless, it is not addressed if clinical trials have already been performed using this target, or “in vivo” studies.

-  MAP kinases, cyclin-dependent kinase 5 and Pin1 are also involved in the kinase cascades related to tau phosphorylation in tauopathies such as Alzheimer´s disease. This aspect should also be addressed along the manuscript (see PMID: 16020737).

-  I am not asking authors to do a second review, just including also these suggestions as a part of the importance of this motor transport machinery in other physiopathological conditions and a more direct application of this great and biochemical review.

Minor point:

-      Lines 77-79: a reference is lacking; the sentence is not well understood since “additional regulators” are not indicated.

Author Response

Thank you for your positive comments and suggestions on the manuscript. We have corrected the manuscript as you suggested.

Major points:

-  Authors focus on the role in cancer of this transcription factor. Nonetheless, it is not addressed if clinical trials have already been performed using this target, or “in vivo” studies.

Reply

We have corrected page 16 in Section 8, as below.

(text before)

Development of chemical agents that inhibit interaction between Hsp90 and Pin1 rather than inhibiting Hap90 ATPase activity could be a promising approach for cancer chemotherapy.

(Revised text)

Development of chemical agents that inhibit Pin1 rather than Hap90 ATPase activity could be a promising approach for cancer chemotherapy (reviewed in [130-132]). For instance, a Pin1 inhibitor all-trans retinoic acid has been used for a therapy of acute promyelocytic leukemia in animal models and human patients [133], and all-trans retinoic acid reduced growth of transplanted tamoxifen-resistant human breast cancer cells in mice [134]. Dubiella et al. [135] showed that a Pin1 inhibitor Sulfopin reduced tumor progression and conferred survival benefit in animal models, whilst Liu et al. [136] developed a delivery system of a Pin1 inhibitor AG177724 targeting to cancer associated fibroblasts and observed the inhibition of tumor growth in mice.

References:

[130] Xu GG, Etzkorn FA. Pin1 as an anticancer drug target. Drug News Perspect. 2009 Sep;22(7):399-407. doi: 10.1358/dnp.2009.22.7.1381751. PMID: 19890497.

[131] Zhou XZ, Lu KP. The isomerase PIN1 controls numerous cancer-driving pathways and is a unique drug target. Nat Rev Cancer. 2016 Jul;16(7):463-78. doi: 10.1038/nrc.2016.49. Epub 2016 Jun 3. PMID: 27256007.

[132] Wu W, Xue X, Chen Y, Zheng N, Wang J. Targeting prolyl isomerase Pin1 as a promising strategy to overcome resistance to cancer therapies. Pharmacol Res. 2022 Oct;184:106456. doi: 10.1016/j.phrs.2022.106456. Epub 2022 Sep 16. PMID: 36116709.

[133] Wei S, Kozono S, Kats L, Nechama M, Li W, Guarnerio J, Luo M, You MH, Yao Y, Kondo A, Hu H, Bozkurt G, Moerke NJ, Cao S, Reschke M, Chen CH, Rego EM, Lo-Coco F, Cantley LC, Lee TH, Wu H, Zhang Y, Pandolfi PP, Zhou XZ, Lu KP. Active Pin1 is a key target of all-trans retinoic acid in acute promyelocytic leukemia and breast cancer. Nat Med. 2015 May;21(5):457-66. doi: 10.1038/nm.3839. Epub 2015 Apr 13. PMID: 25849135; PMCID: PMC4425616.

[134] Huang S, Chen Y, Liang ZM, Li NN, Liu Y, Zhu Y, Liao D, Zhou XZ, Lu KP, Yao Y, Luo ML. Targeting Pin1 by All-Trans Retinoic Acid (ATRA) Overcomes Tamoxifen Resistance in Breast Cancer via Multifactorial Mechanisms. Front Cell Dev Biol. 2019 Dec 6;7:322. doi: 10.3389/fcell.2019.00322. PMID: 31867329; PMCID: PMC6908472.

[135] Dubiella C, Pinch BJ, Koikawa K, Zaidman D, Poon E, Manz TD, Nabet B, He S, Resnick E, Rogel A, Langer EM, Daniel CJ, Seo HS, Chen Y, Adelmant G, Sharifzadeh S, Ficarro SB, Jamin Y, Martins da Costa B, Zimmerman MW, Lian X, Kibe S, Kozono S, Doctor ZM, Browne CM, Yang A, Stoler-Barak L, Shah RB, Vangos NE, Geffken EA, Oren R, Koide E, Sidi S, Shulman Z, Wang C, Marto JA, Dhe-Paganon S, Look T, Zhou XZ, Lu KP, Sears RC, Chesler L, Gray NS, London N. Sulfopin is a covalent inhibitor of Pin1 that blocks Myc-driven tumors in vivo. Nat Chem Biol. 2021 Sep;17(9):954-963. doi: 10.1038/s41589-021-00786-7. Epub 2021 May 10. PMID: 33972797; PMCID: PMC9119696.

[136] Liu J, Wang Y, Mu C, Li M, Li K, Li S, Wu W, Du L, Zhang X, Li C, Peng W, Shen J, Liu Y, Yang D, Zhang K, Ning Q, Fu X, Zeng Y, Ni Y, Zhou Z, Liu Y, Hu Y, Zheng X, Wen T, Li Z, Liu Y. Pancreatic tumor eradication via selective Pin1 inhibition in cancer-associated fibroblasts and T lymphocytes engagement. Nat Commun. 2022 Jul 25;13(1):4308. doi: 10.1038/s41467-022-31928-7. PMID: 35879297; PMCID: PMC9314377.

-  MAP kinases, cyclin-dependent kinase 5 and Pin1 are also involved in the kinase cascades related to tau phosphorylation in tauopathies such as Alzheimer´s disease. This aspect should also be addressed along the manuscript (see PMID: 16020737).

Reply: The last paragraph in p17 has been changed as shown below.

In summary, we propose that ERK, JNK and Cdk5 control Nrf2 phosphorylation inducing a formation of multiprotein complex containing Hsp90α-p23-Pin1-Nrf2-importins to associate with dynein motors to move from cell membrane signaling compartments toward nuclear pores along microtubules (Fig. 3 C). Therefore, integrity of microtubules is important to ensure nuclear translocation of Nrf2 and other signaling molecules. It is important to note that tau family proteins control the assembly and maintenance of the structural stability of microtubules and that proline-directed kinases, MAP kinases, GSK-3β and Cdk5, can phosphorylate tau proteins and affect their functions (reviewed in [148-150]). GSK-3β phosphorylated tau reduces the affinity to microtubules causing disfunction of microtubules [151-152]. Interestingly, Pin1 interacts with phosphorylated tau proteins and restores the ability of tau to bind microtubules and promote assembly in vitro [153], and downregulates GSK-3β-mediated phosphorylation of tau [154,155]. These studies suggest that Pin1/tau axis is also important for microtubule-mediated facilitated translocation of Nrf2 to the nucleus. As tau controls axonal transport and neurite outgrowth in neurons, defects in tau function could lead to neurodegeneration [156]. Thus, Pin1 plays an important but opposite role in the pathogenesis of Alzheimer’s disease and many human cancers (reviewed in [157,158]).

References:

[148] Mandelkow EM, Biernat J, Drewes G, Gustke N, Trinczek B, Mandelkow E. Tau domains, phosphorylation, and interactions with microtubules. Neurobiol Aging. 1995 May-Jun;16(3):355-62; discussion 362-3. doi: 10.1016/0197-4580(95)00025-a. PMID: 7566345.

[149] Johnson GV, Stoothoff WH. Tau phosphorylation in neuronal cell function and dysfunction. J Cell Sci. 2004 Nov 15;117(Pt 24):5721-9. doi: 10.1242/jcs.01558. PMID: 15537830.

[150] Dehmelt L, Halpain S. The MAP2/Tau family of microtubule-associated proteins. Genome Biol. 2005;6(1):204. doi: 10.1186/gb-2004-6-1-204. Epub 2004 Dec 23. PMID: 15642108; PMCID: PMC549057.

[151] Wagner U, Utton M, Gallo JM, Miller CC. Cellular phosphorylation of tau by GSK-3 beta influences tau binding to microtubules and microtubule organisation. J Cell Sci. 1996 Jun;109 ( Pt 6):1537-43. doi: 10.1242/jcs.109.6.1537. PMID: 8799840.

[152] Lovestone S, Hartley CL, Pearce J, Anderton BH. Phosphorylation of tau by glycogen synthase kinase-3 beta in intact mammalian cells: the effects on the organization and stability of microtubules. Neuroscience. 1996 Aug;73(4):1145-57. doi: 10.1016/0306-4522(96)00126-1. PMID: 8809831.

[153] Lu PJ, Wulf G, Zhou XZ, Davies P, Lu KP. The prolyl isomerase Pin1 restores the function of Alzheimer-associated phosphorylated tau protein. Nature. 1999 Jun 24;399(6738):784-8. doi: 10.1038/21650. PMID: 10391244.

[154] Min SH, Cho JS, Oh JH, Shim SB, Hwang DY, Lee SH, Jee SW, Lim HJ, Kim MY, Sheen YY, Lee SH, Kim YK. Tau and GSK3beta dephosphorylations are required for regulating Pin1 phosphorylation. Neurochem Res. 2005 Aug;30(8):955-61. doi: 10.1007/s11064-005-6177-0. PMID: 16258844.

[155] Ma SL, Pastorino L, Zhou XZ, Lu KP. Prolyl isomerase Pin1 promotes amyloid precursor protein (APP) turnover by inhibiting glycogen synthase kinase-3β (GSK3β) activity: novel mechanism for Pin1 to protect against Alzheimer disease. J Biol Chem. 2012 Mar 2;287(10):6969-73. doi: 10.1074/jbc.C111.298596. Epub 2011 Dec 19. PMID: 22184106; PMCID: PMC3293570.

[156] Santacruz K, Lewis J, Spires T, Paulson J, Kotilinek L, Ingelsson M, Guimaraes A, DeTure M, Ramsden M, McGowan E, Forster C, Yue M, Orne J, Janus C, Mariash A, Kuskowski M, Hyman B, Hutton M, Ashe KH. Tau suppression in a neurodegenerative mouse model improves memory function. Science. 2005 Jul 15;309(5733):476-81. doi: 10.1126/science.1113694. PMID: 16020737; PMCID: PMC1574647.

[157] Driver JA, Zhou XZ, Lu KP. Pin1 dysregulation helps to explain the inverse association between cancer and Alzheimer's disease. Biochim Biophys Acta. 2015 Oct;1850(10):2069-76. doi: 10.1016/j.bbagen.2014.12.025. Epub 2015 Jan 10. PMID: 25583562; PMCID: PMC4499009.

[158] Lanni C, Masi M, Racchi M, Govoni S. Cancer and Alzheimer's disease inverse relationship: an age-associated diverging derailment of shared pathways. Mol Psychiatry. 2021 Jan;26(1):280-295. doi: 10.1038/s41380-020-0760-2. Epub 2020 May 7. PMID: 32382138.

Minor point:

-      Lines 77-79: a reference is lacking; the sentence is not well understood since “additional regulators” are not indicated.

Reply: The sentence was changed as shown below.

(text before)

However, increased Nrf2 levels in the cytoplasm does not automatically lead to nuclear accumulation of Nrf2, as transport of Nrf2 into the nucleus is tightly controlled requiring additional regulators (Fig. 1 A).

(Revised text)

However, increased Nrf2 levels in the cytoplasm do not automatically lead to nuclear accumulation of Nrf2, as transport of Nrf2 into the nucleus is tightly controlled requiring additional regulators such as importins [16].

[16] [20] Theodore M, Kawai Y, Yang J, Kleshchenko Y, Reddy SP, Villalta F, Arinze IJ. Multiple nuclear localization signals function in the nuclear import of the transcription factor Nrf2. J Biol Chem. 2008 Apr 4;283(14):8984-94. doi: 10.1074/jbc.M709040200. Epub 2008 Jan 31. Erratum in: J Biol Chem. 2008 May 16;283(20):14176. PMID: 18238777; PMCID: PMC2276363.

Reviewer 3 Report

The review titled “Stress activated MAP kinases and cyclin-dependent kinase 5 mediate nuclear translocation of Nrf2 via Hsp90α-Pin1-dynein motor transport machinery”, reports important information and points of view of the authors that arouse considerable interest.

A detailed analysis of the literature was reported and data were well organized and well illustrated. The tables showing the cell types, activators and effects, accompanied by bibliographic citations, are very informative. Even the figures are well coordinated with the rest of the text. The conclusions reported by the authors are consistent with the cited literature.

In conclusion, the manuscript is well structured and data are well illustrated and merits to be published on Antioxidants journal.

Author Response

Thank you for your positive comments.

Round 2

Reviewer 2 Report

I would like to thank the authors for taking my suggestions into account. Congratulations for this paper